# Natural Product-Based Glycolysis Inhibitors as a Therapeutic Strategy for Epidermal Growth Factor Receptor–Tyrosine Kinase Inhibitor-Resistant Non-Small Cell Lung Cancer

**DOI:** 10.3390/ijms25020807

**Published:** 2024-01-09

**Authors:** Wonyoung Park, Jung Ho Han, Shibo Wei, Eun-Sun Yang, Se-Yun Cheon, Sung-Jin Bae, Dongryeol Ryu, Hwan-Suck Chung, Ki-Tae Ha

**Affiliations:** 1Department of Korean Medical Science, School of Korean Medicine, Pusan National University, Yangsan 50612, Republic of Korea; jinling0122@pusan.ac.kr; 2Korean Medical Research Center for Healthy Aging, Pusan National University, Yangsan 50612, Republic of Korea; yanges@pusan.ac.kr (E.-S.Y.); chunsay1008@pusan.ac.kr (S.-Y.C.); 3Korean Medicine Application Center, Korea Institute of Oriental Medicine, Daegu 41062, Republic of Korea; hanjh1013@kiom.re.kr; 4Department of Molecular Cell Biology, School of Medicine, Sungkyunkwan University, Suwon 16419, Republic of Korea; weishibo8186@163.com; 5Department of Molecular Biology and Immunology, Kosin University College of Medicine, Busan 49267, Republic of Korea; dr.baesj@kosin.ac.kr; 6Department of Biomedical Science and Engineering, Gwangju Institute of Science and Technology, Gwangju 61005, Republic of Korea; dryu@gist.ac.kr

**Keywords:** natural products, NSCLC, EGFR-TKI, glycolysis inhibitor, PDK, LDHA

## Abstract

Non-small cell lung cancer (NSCLC) is a leading cause of cancer-related deaths worldwide. Targeted therapy against the epidermal growth factor receptor (EGFR) is a promising treatment approach for NSCLC. However, resistance to EGFR tyrosine kinase inhibitors (TKIs) remains a major challenge in its clinical management. EGFR mutation elevates the expression of hypoxia-inducible factor-1 alpha to upregulate the production of glycolytic enzymes, increasing glycolysis and tumor resistance. The inhibition of glycolysis can be a potential strategy for overcoming EGFR-TKI resistance and enhancing the effectiveness of EGFR-TKIs. In this review, we specifically explored the effectiveness of pyruvate dehydrogenase kinase inhibitors and lactate dehydrogenase A inhibitors in combating EGFR-TKI resistance. The aim was to summarize the effects of these natural products in preclinical NSCLC models to provide a comprehensive understanding of the potential therapeutic effects. The study findings suggest that natural products can be promising inhibitors of glycolytic enzymes for the treatment of EGFR-TKI-resistant NSCLC. Further investigations through preclinical and clinical studies are required to validate the efficacy of natural product-based glycolytic inhibitors as innovative therapeutic modalities for NSCLC.

## 1. Introduction

Lung cancer is the leading cause of cancer-related deaths, with approximately 1.8 million deaths worldwide in 2020 [1]. Non-small cell lung cancer (NSCLC; 82%) and small cell lung cancer (SCLC; 14%) account for the majority of cases of lung cancer [2,3]. Overall, NSCLC is the most frequent type of lung cancer. Based on the histological characteristics, NSCLC is further classified into lung adenocarcinoma (50–60%), squamous cell carcinoma (20–30%), and large cell carcinoma (10–20%) [4].

The epidermal growth factor receptor (EGFR) is one of the most common driving mutations in NSCLC [5]. *EGFR* mutations are more common in Asian populations (approximately 50%) than in populations from the United States and Europe (approximately 10%) [6]. Consequently, *EGFR* is one of the most significant targetable mutations in NSCLC and is widely explored in cancer research, medication development, and diagnosis. A decade ago, the average survival of patients with advanced NSCLC and *EGFR* mutations was less than 2 years [7]. Currently, patients receiving third-generation EGFR tyrosine kinase inhibitor (TKI) treatment have a median survival time of more than 3 years [8]. A third-generation EGFR-TKI (osimertinib) has good treatment efficacy but is also associated with the development of secondary resistance [9]. Therefore, overcoming EGFR-TKI resistance is important.

Emerging evidence highlights the correlation between glycolysis and drug resistance in cancer. Within cancer cells, glycolysis-related enzymes play pivotal roles in enhancing resistance to chemotherapy [10]. Pyruvate dehydrogenase kinase (PDK) is a key enzyme that is involved in the regulation of glucose metabolism and is frequently overexpressed in cancer cells, resulting in increased glycolysis and lactate production [11]. Moreover, PDK inhibition has been demonstrated to reduce cancer cell growth and enhance susceptibility to chemotherapy [12]. Lactate dehydrogenase (LDH) regulates the conversion and production of pyruvate and lactic acid [13]. Inhibition of LDH A (LDHA) increases oxidative stress and reduces chemoresistance [14]. Therefore, targeting glycolytic enzymes such as PDK or LDHA may be a promising strategy to overcome EGFR-TKI resistance in patients with NSCLC.

Natural products have the potential to be effective as therapeutic agents because of their lower toxicity and higher specificity than synthetic compounds [15]. A growing interest has been observed in identifying glycolysis inhibitors from natural sources that can be used as safe and effective alternatives [16,17]. These inhibitors have demonstrated promising results in preclinical models [18,19,20]. Therefore, we compiled natural products-derived inhibitors of enzymes that are involved in glycolysis, including PDKs and LDHA, with the aim of understanding the potential of these compounds in overcoming EGFR-TKI resistance.

This review provides an overview of the current understanding of the role of targeted therapy in NSCLC, delves into the historical processes underlying EGFR-TKI resistance, and explores the potential of utilizing glycolysis inhibitors that are derived from natural products as a strategy for overcoming EGFR-TKI resistance.

## 2. Targeted Therapy in NSCLC

The treatment for NSCLC is multifaceted and can be tailored to meet the needs of individual patients. Surgical interventions, including lobectomy and wedge resection, are designed to surgically remove tumors and associated lymph nodes. Radiation therapy utilizes high-energy X-rays to target cancer cells, and chemotherapy employs drugs such as carboplatin and cisplatin to disrupt the growth of these cells. A pivotal shift in the treatment paradigm has occurred with the identification of specific targetable mutations in patients with advanced NSCLC [21], and the use of drugs that are designed for these genetic mutations or altered proteins that promote cancer cell growth and spread [22]. The targeted therapies include those directed towards *EGFR*, anaplastic lymphoma kinase (*ALK*), ROS proto-oncogene 1, receptor tyrosine kinase (*ROS1*), v-raf murine sarcoma viral oncogene homolog B1 (*BRAF*), proto-oncogene, receptor tyrosine kinase (*MET*), RET proto-oncogene (*RET*), Kirsten rat sarcoma virus (*KRAS*), and programmed death-ligand 1 (*PD-L1*), and they work by disrupting the signaling pathways that are responsible for cancer cell growth [22]. Unlike conventional treatments, targeted therapy allows for a more precise and personalized approach which enhances treatment efficacy while minimizing side effects and provides a valuable alternative for patients who may not respond well to standard therapies. Additionally, targeted therapy plays a key role in overcoming resistance to traditional treatments and recognizes the unique genetic profile of each patient with cancer [23]. The integration of targeted therapy into NSCLC treatment strategies represents a significant advancement in improving patient outcomes and underscores the necessity for a nuanced and tailored therapeutic approach in the era of precision medicine.

The efficacy of targeted therapies for NSCLC varies depending on individual patient characteristics and genetic testing results. To determine the best therapeutic options for individual patients, NSCLC cells are subjected to molecular profiling. In 2022, the National Comprehensive Cancer Network (NCCN) expanded its guidelines for metastatic NSCLC to include “broad molecular profiles including *EGFR* (Figure 1A), *ALK*, *HER2*, *MET*, *NTRK*, *RET*, *ROS1* (Figure 1B), *KRAS*, *BRAF* (Figure 1C), and *PD-L1* (Figure 1D) [24]. The NCCN guidelines recommend several Food and Drug Administration (FDA)-approved targeted therapeutic agents as first-line therapies for patients with specific mutations. Afatinib [25], dacomitinib [26], erlotinib [27], gefitinib [28], and osimertinib [29] are utilized for patients with the *EGFR* mutation; amivantamab-vmjw [30] and mobocertinib are employed for patients with *EGFR* exon 20 insertions; targeted therapy for other receptor tyrosine kinases (ALK, HER2, MET, NTRK, RET, ROS1) include alectinib [31], brigatinib [32], capmatinib [33], ceritinib [34], crizotinib [35], entrectinib [36], larotrectinib [37], lorlatinib [38], pralsetinib [39], selpercatinib [40], and tepotinib [41] (Figure 2). Sotorasib [42] has been approved for *KRAS* mutations, and dabrafenib [43] and trametinib [43] have been approved for *BRAF* mutations. Immunotherapy targeting PD-L1 is also considered a form of targeted therapy. The FDA has approved atezolizumab [44], bevacizumab [45], ipilimumab [46], nivolumab [47], and pembrolizumab [48] as PD-L1 targeting therapies. Monoclonal antibodies are a type of targeted therapy which were initially developed as a cancer therapy targeting EGFR [49]. Necitumumab, an EGFR monoclonal antibody, has been approved as a combination therapy for metastatic squamous NSCLC [50].

## 3. EGFR-TKIs in NSCLC Treatment

EGFR, an oncogenic receptor tyrosine kinase (TK) belonging to the ErbB receptor family, is activated upon binding to specific ligands, including epidermal growth factor (EGF) (Figure 3) [51]. Several isotypic ErbB family receptors, such as human epidermal growth factor receptor (HER) 2, HER3, and HER4, play key roles in the development of NSCLC. In normal cells, EGFR activation leads to receptor homo- or hetero-dimerization and autophosphorylation of the intracellular TK domain, which in turn activates signaling pathways that regulate cellular proliferation, migration, and differentiation [52,53]. However, *EGFR* is frequently altered in tumor cells, and these alterations can lead to abnormal signaling, resulting in cancer cell proliferation, invasion, and metastasis [54]. To attenuate the effects of *EGFR* mutation-induced aberrant signaling, EGFR-TKIs have been developed to inhibit enzymatic activity by binding to the TK domain of EGFRs [55].

Mutations in kinase-activating *EGFRs* and overexpression of the EGFR protein are the predominant changes observed in cancer (Figure 1A) [60,61]. The most common *EGFR*-activating mutations include an in-frame deletion in exon 19 within codons 746–750 (19D; 45–50%) and a single-base substitution of arginine with leucine at codon 858 in exon 21 (L858R; approximately 35–45%) near the adenosine triphosphate (ATP)-binding pocket of the TK domain [54,62]. First-generation EGFR-TKIs (gefitinib and erlotinib), which reversibly bind to the ATP-binding site of the *EGFR* tyrosine kinase domain, have resulted in considerable improvements in the outcome for patients with *EGFR*-mutated NSCLC (L858R and 19D) (Table 1) [63,64]. Additionally, less common *EGFR* mutations such as G719X, L861Q, and S768I have demonstrated responsiveness to first-generation EGFR-TKI treatment [65,66]. According to a recent study, patients who received subsequent EGFR-TKI treatment lived the longest, with a median overall survival (OS) of 31.3 months (95% confidence interval (CI), 23.9–38.7 months) compared to those who received chemotherapy (median OS, 19.4 months; 95% CI, 18.5–20.3 months) or no subsequent treatment (median OS, 2.4 months; 95% CI, 1.3–3.5 months) [67]. This study suggests that patients with *EGFR*-mutated NSCLC may benefit from further treatment with EGFR-TKIs. However, after treatment with gefitinib, erlotinib, or afatinib (second-generation EGFR-TKIs) for approximately 9–14 months, up to 50% of the patients developed T790M-mediated resistance [68,69]. In comparison to gefitinib, dacomitinib, a second-generation EGFR-TKI, has significantly improved progression-free survival in the first-line treatment of patients with *EGFR*-mutation-positive NSCLC; however, the drug also has the potential to directly induce secondary mutations such as T790M [70,71]. Osimertinib is a third-generation EGFR-TKI that binds to the C797 residue in the ATP-binding site of *EGFR* and exhibits high selectivity for both *EGFR*-activating mutations and the secondary acquired *EGFR* T790M mutation [9]. Patients treated with osimertinib had a median OS of 38.6 months (95% CI, 34.5–41.8), whereas those in the comparator group had a median OS of 31.8 months [8]. Osimertinib is a promising third-generation EGFR-TKI, and its combination with platinum-based chemotherapy may provide additional treatment options for *EGFR*-mutated NSCLC [72]. Although the development of targeted therapies including the first-generation (gefitinib and erlotinib), second-generation (afatinib and dacomitinib), and third-generation (osimertinib) EGFR-TKIs has demonstrated substantial improvements in the overall survival of patients with *EGFR*-mutated NSCLC, the emergence of secondary resistance is challenging [73]. Thus, continued research and exploration of novel therapeutic strategies are imperative to address and overcome these resistance mechanisms and ensure sustained efficacy and prolonged benefits for patients with *EGFR*-mutated NSCLC.

## 4. Enhanced Glycolysis in EGFR-TKI-Resistant NSCLC

Glycolysis is a metabolic pathway that converts glucose to lactate. Moreover, glycolysis is orchestrated by a series of glycolytic enzymes, and the dysregulation of the procedure has been implicated in conferring resistance to EGFR-targeted therapies in cancer cells [79]. The cancer cells often favor glycolysis (the Warburg effect), despite the presence of oxygen [80]. The Warburg effect suggests that cancer cells favor glycolysis even when oxygen is available, because glycolysis is advantageous for their rapid division [81]. Although glycolysis is less efficient in terms of energy production, the process allows cancer cells to generate energy quickly and provides essential biosynthetic precursors for the synthesis of various cellular components that are required for cell growth and division [82]. However, the production of free radicals during glycolysis poses a potential challenge, as these reactive oxygen species (ROS) can have damaging effects on cellular components including deoxyribonucleic acid (DNA), proteins, and lipids [83]. However, the NSCLC cells exhibit several adaptive mechanisms for managing proliferation despite oxidative stress. Cells in NSCLC employ a combination of antioxidant defenses and survival signaling, including the phosphatidylinositol 3-kinase (PI3K)/AKT pathway, metabolic adaptations, DNA repair mechanisms, and adaptation to hypoxic conditions, to manage and sustain proliferation in the presence of oxidative stress during glycolysis [84,85,86,87]. The heightened glycolysis and increased lactate production observed in cancer cells significantly contribute to the development of resistance to EGFR-TKIs [88]. Acknowledging that the complex interplay between various glycolytic enzymes plays a pivotal role in mediating resistance to EGFR-TKIs is crucial. A previous study proposed that elevated glycolytic activity could be predictive of gefitinib resistance in patients with *EGFR*-mutant NSCLC receiving first-line gefitinib treatment [89]. This suggests that targeting the glycolytic pathway and its associated enzymes can be a promising avenue for novel therapeutic approaches aimed at overcoming EGFR-TKI resistance.

In EGFR-TKI-resistant NSCLC cells, increased glucose uptake is primarily facilitated by the upregulation of glucose transporter 1 (GLUT1), which is a critical regulator of glucose entry into cells [90]. This increase in GLUT1 enhances the efficiency of glucose transport across the cell membrane, ensuring a constant supply of glucose to fuel glycolysis (the preferred energy-producing pathway in resistant cells) [91]. A previous study discovered that inhibiting glycolysis using 2-deoxy-d-glucose enhanced sensitivity to afatinib (a second-generation irreversible EGFR-TK) in NSCLC cells with acquired resistance due to the secondary EGFR T790M mutation [92]. Hexokinase plays an important role in the early stages of glycolysis by catalyzing glucose phosphorylation, which is the first step of this metabolic pathway [93]. Additionally, the inhibition of hexokinase 2 (HK2) sensitizes resistant NSCLC cells to gefitinib. This is suggestive of an important role of HK2 in the development of resistance mechanisms [90]. Pyruvate kinase M2 (PKM2) is a crucial enzyme that regulates the final step of the glycolytic pathway and facilitates the transformation of phosphoenolpyruvate into pyruvate [94]. Additionally, PKM2 can translocate to the nucleus and activate the signal transducer and activator of transcription 3, which can cause resistance to gefitinib [95]. Pyruvate is a critical metabolite in cellular metabolism and is involved in several significant metabolic pathways depending on the cellular context and environmental factors [94]. Pyruvate regulation is complex and involves key enzymes, particularly PDK and LDHA. PDK governs the entry of pyruvate into the citric acid cycle and glycolysis [96], whereas LDHA catalyzes the conversion of pyruvate to lactate under anaerobic conditions [97]. Notably, PDK and LDHA are significantly associated with EGFR-TKI resistance. The interaction between PDK and LDHA will be explored in future studies.

The intricate regulation of glycolysis is a critical factor in EGFR-TKI resistance in NSCLC [98]. Targeting key players of the glycolytic pathway, such as GLUT1, HK2, PKM2, PDK, and LDHA, is a promising avenue for novel therapeutic strategies aimed at overcoming EGFR-TKI resistance [90,95,99,100,101]. Building on this understanding, the following section describes the exploration of glycolytic inhibitors that are derived from natural products and provides insights into potential nature-inspired interventions to disrupt this crucial metabolic pathway and enhance the efficacy of EGFR-targeted therapies.

## 5. Advantages of PDK Inhibition against EGFR-TKI Resistance and Inhibitors from Natural Products

PDK regulates the activity of the pyruvate dehydrogenase complex (PDC), which converts pyruvate to acetyl-CoA for mitochondrial ATP production [102]. PDK inhibits PDC activity via phosphorylation, resulting in a decreased conversion of pyruvate to acetyl-CoA. Instead, pyruvate is diverted toward lactate production via glycolysis [103]. Cancer cells exhibit high levels of PDK1 expression and activity, which promotes glycolysis and facilitates cancer cell survival and proliferation [104]. The induction of PDK by hypoxia-inducible factor-1 alpha (HIF-1α) has been demonstrated to cause chemotherapy resistance in cancer cells [12]. HIF-1α-induced upregulation of PDK1 inhibits PDC activity, causing a shift in the cancer cell metabolism towards anaerobic glycolysis, and decreases the production of ROS [105]. Consequently, cancer cells become resistant to chemotherapeutic drugs that rely on the cytotoxic effects of ROS [106]. In addition, the shift towards anaerobic glycolysis provides cancer cells with a metabolic advantage, allowing them to survive and proliferate in conditions with limited oxygen and nutrients [79]. Inhibiting PDK allows more pyruvate to enter the mitochondria, promoting oxidative phosphorylation (OXPHOS) and enhancing the production of ROS. This, in turn, causes oxidative stress, damages cellular components, and triggers apoptotic pathways, ultimately leading to cancer cell death [18,19,20,107]. Crystal structure studies have indicated that the pyruvate-binding domain (located at the N-terminal regulatory domain), the lipoamide-binding domain, and the nucleotide-binding domain (located at the C-terminal catalytic domain) are all critical for controlling PDK activity [108]. Dichloroacetate (DCA) is an orally available small-molecule PDK inhibitor that shifts the cancer cell metabolism from glycolysis to OXPHOS by inhibiting PDK activity [109]. In a previous study, the combination of DCA with the first-generation EGFR-TKIs erlotinib and gefitinib dramatically reduced the viability of *EGFR*-mutant NSCLC cells (NCI-H1975 and NCI-H1650) [110]. Another study suggested that DCA in combination with rociletinib (a third-generation EGFR-TKI) along with radiation therapy might be a promising therapeutic strategy for treating NSCLC [111].

Several natural products have been reported to have PDK-inhibiting activity, for example, huzhangoside A isolated from *Anemone rivularis*, ilimaquinone isolated from *Smenospongia cerebriformis*, and hemistepsin A isolated from, *Hemistepta lyrate* (Table 2) [18,19,20]. Although the precise IC_50_ values for the PDK enzyme activity of these natural products have not been reported, their anticancer effects have been confirmed using in vitro and/or in vivo studies of colon, lung, breast, and liver cancers. Dicoumarol from *Melilotus officinalis*, cryptotanshinone from *Salvia miltiorrhiza*, and quercetin from various fruits and vegetables exhibit inhibitory effects on PDK and have been demonstrated to have anticancer activities against hepatocellular carcinoma, pancreatic cancer, and lung cancer [112,113,114]. In addition, several natural products, such as baicalin, β-asarone, betulinic acid, cardamonin, and helichrysetin, are known to inhibit the expression of PDK1 [115,116,117,118,119]. Most of the substances mentioned above inhibit the upstream factors such as cellular-myelocytomatosis oncogene (c-Myc), HIF-1α, and phosphatase and tensin homolog/Akt that regulate the expression of PDK1. Although these natural products are not direct inhibitors of PDK1, they can inhibit glycolysis; therefore, they may exert effects that are similar to those of synthetic PDK inhibitors. Further clinical trials are needed to determine the efficacy and safety of these PDK inhibitors.

Therefore, PDK inhibition may be a useful therapeutic strategy for overcoming EGFR-TKI resistance. Nevertheless, additional studies are necessary to substantiate the mechanisms and clinical efficacy of PDK inhibitors in the treatment of EGFR-TKI-resistant NSCLC.

## 6. Natural Product-Derived LDHA Inhibitors and Their Advantage against EGFR-TKI Resistance

LDH plays vital roles in cellular respiration [123] by converting lactate to pyruvate or pyruvate to lactate, thereby maintaining an equilibrium between Nicotinamide adenine dinucleotide (NAD^+^) and its reduced form (NADH), which are essential elements in energy production [124,125]. In humans, LDH utilizes His193 as a proton acceptor and collaborates with coenzyme-binding residues (Arg99 and Asn138) and substrate-binding (Arg106, Arg169, and Thr248) residues [126]. Two main types of subunits are present in LDH, denoted as M (for muscle) and H (for heart) and encoded by genes—*LDH-A* and *LDH-B*, respectively [127]. The combination of these M (LDHA) and H (LDHB) subunits leads to the formation of tetrameric LDH of varying compositions (e.g., LDH-1, LDH-2, LDH-3, LDH-4, and LDH-5), each with different kinetic properties and tissue distributions [127]. Furthermore, LDHA is the most abundant isotype in the skeletal muscle and efficiently catalyzes the conversion of pyruvate to lactate and NADH to NAD^+^ [128]. In contrast, LDHB is predominantly present in the heart, liver, and brain, where it facilitates the conversion of lactate to pyruvate and NAD^+^ to NADH [127,129,130].

Abnormal LDH activity has been associated with a range of diseases including cancer, metabolic disorders, neurodegenerative diseases, and cardiovascular diseases [131,132,133,134]. In cancers, dysregulated LDH activity influences tumor progression by promoting the Warburg effect [135]. The conversion of pyruvate to lactate, favored by LDHA overexpression, partially contributes to the acidification of the tumor microenvironment [136]. This acidification is linked to the progression and metastasis of cancer and other diseases [79]. Elevated levels of plasma LDH can be used as a prognostic factor in patients with *EGFR*-mutated NSCLC [137,138]. However, LDHA inhibition may overcome EGFR TKI resistance.

Several LDHA inhibitors have been derived from natural products (Table 3). Apigenin has been reported to reduce LDHA messenger ribonucleic acid (RNA) expression [139]. Berberine improves ischemia/reperfusion injury by downregulating LDHA activity and subsequently decreases lactate production [140]. Capsaicin suppresses the EGF-induced invasion and migration of human fibrosarcoma cells [141]. In our previous study, catechin, which is known to enhance cardiovascular health and reduce oxidative stress and inflammation [142], exhibited a potent inhibitory effect on LDHA by directly binding to the Thr94, Ala95, Gln99, Arg105, Ser136, Arg168, His192, and Thr247 residues of LDHA [14]. Curcumin, extracted from *Curcuma longa*, inhibits glycolysis by downregulating the expression of HK2 and LDHA, thereby inducing mitochondria-mediated apoptosis in colorectal cancer cells [143]. Curcumin has been reported to overcome the resistance to EGFR-TKI (gefitinib and erlotinib) [144,145]. Epigallocatechin gallate (EGCG), a significant biologically active component of green tea, has been identified to have LDHA-inhibitory activity [146]. EGCG has a synergistic effect when used in combination with EGFR-TKI for head and neck cancer [147,148]. LDHA is a single-stranded DNA-binding protein that stimulates cell transcription [149]. Galloflavin acts as an inhibitor of LDHA, preventing it from binding to single-stranded DNA and reducing RNA production [150]. Leonurine (known for its cardioprotective effect) has been demonstrated to reduce LDH activity and has antioxidant properties [151]. Quercetin, extracted from *Quercus*, can decrease the activity and expression of LDHA, suppress PI3K/AKT signaling, and regress Dalton’s lymphoma growth [152]. Ursolic acid has antioxidant, antidiabetic, antibacterial, and anticancer effects [153] and can suppress LDHA expression in breast cancer [154].

The diverse range of natural LDHA inhibitors highlighted in this discussion, such as apigenin, berberine, capsaicin, catechin, curcumin, EGCG, galloflavin, leonurine, quercetin, and ursolic acid, demonstrate the potential of natural compounds to modulate LDHA activity and associated pathologies (Table 3). Curcumin and EGCG have emerged as promising candidates demonstrating efficacy as LDHA inhibitors and for overcoming EGFR-TKI resistance, particularly for cancer treatment. The interplay between LDHA and EGFR-TKI resistance presents a fascinating avenue for further investigation. Though the current research on natural remedy treatment options is relatively limited, exploring new options holds promise for advancing our understanding and refining treatment strategies.

## 7. Natural Products Suppressing Other Glycolytic Enzymes and Their Use for EGFR-TKI Resistance

Cucurbitacin D, a naturally occurring compound known for its anticancer properties, can impede the growth, invasion, and metastasis of prostate cancer cells by orchestrating the reprogramming of their glucose metabolism network [155]. Cucurbitacin D accomplishes this by binding to GLUT1, a membrane protein that facilitates glucose entry into cells and inhibits cell function, consequently diminishing glucose uptake by prostate cancer cells. Additionally, cucurbitacin D hampers ATP production in these cells, leading to apoptosis. Cucurbitacin B, another member of the cucurbitacin family, has exhibited therapeutic potential when combined with the EGFR-TKI gefitinib [156]. Genistein is a natural isoflavone that can directly downregulate HIF-1α, thereby inactivating GLUT1 and HK2 to suppress aerobic glycolysis [157]. α-Hederin, a pentacyclic triterpenoid saponin that is present in the leaves of *Hedera helix*, is known for anti-inflammatory, antioxidant, and anticancer properties [158]. Moreover, α-Hederin inhibits the growth of lung cancer cell lines (A549, NCI-H460, and NCI-H292) by suppressing glycolysis-related factors including GLUT1, PKM2, LDHA, and HK2 proteins and demonstrates efficacy in inhibiting tumor growth in an A549-injected mouse model [159]. β-elemene is a natural compound that displays antimetastatic efficacy by blocking PKM2 transformation and nuclear translocation [160]. β-elemene can overcome gefitinib resistance by inducing fructose-1,6-bisphosphatase [161]. Licochalcone, a natural compound from *Glycyrrhiza uralensis*, is a potent HIF-1α inhibitor that can suppress the expression of GLUT1 and PDK1 by inhibiting HIF-1α in HCT116 cells [162]. Additionally, licochalcone-A can overcome mesenchymal-epithelial transition factor (c-Met) overexpression-mediated gefitinib resistance by promoting c-Met ubiquitination [163]. Tanshinone IIA is a natural product extracted from *Salvia miltiorrhiza Bunge* [164]. Tanshinone IIA inhibits the development and proliferation of oral squamous cell carcinoma cells by suppressing Akt-c-Myc signaling and HK2-mediated glycolysis by diminishing HK2 expression at the transcriptional level [165]. Tanshinone IIA is an EGFR inhibitor that suppresses the growth of NSCLC cells by targeting the EGFR-Akt-myeloid cell leukemia-1 axis. Sulforaphane can modulate HIF-1α stability in human colon cancer cells [166] and downregulates glycolytic enzymes, including HK2, PKM2, and PDH, in bladder cancer [167]. In EGFR-TKI-resistant NSCLC cells, SFN treatment reduces EGFR expression and inhibits tumor growth [168]. The anticancer effects of EGCG have been attributed to multiple molecular mechanisms. EGCG inhibits HK2 expression and induces apoptosis in human tongue carcinomas [169]. Furthermore, EGCG disrupts the binding of EGF to EGFR, leading to the inhibition of EGFR TK activity [170,171]. EGCG can also induce the internalization of EGFR into endosomes, rendering it inaccessible to EGF ligands [172]. In another study, EGCG was demonstrated to overcome gefitinib resistance by inhibiting autophagy and enhancing cell death by targeting the extracellular signal-regulated kinase pathway in NSCLC [173]. Shikonin has been detected in *lithospermum erythrorhizon* and identified as a glycolysis inhibitor that suppresses PKM2 [174]. Shikonin exhibits a synergistic anticancer effect when combined with gefitinib via putative molecular processes that are associated with PKM2/STAT3/cyclinD1 inhibition.

In summary, a range of natural compounds such as cucurbitacin D, cucurbitacin B, genistein, α-Hederin, β-elemene, licochalcone, tanshinone IIA, sulforaphane, EGCG, and shikonin demonstrate promising glycolysis-inhibiting effects. These compounds target various glycolytic components and represent potential therapeutic avenues for overcoming EGFR-TKI resistance and inhibiting cancer cell growth (Table 4).

## 8. Perspectives and Conclusions

Drug resistance is a major challenge in cancer treatment, particularly in targeted therapy [175]. Cancer cells can evolve and adapt to the effects of targeted therapies, thereby reducing treatment effectiveness over time. Resistance to drugs can occur through mechanisms such as mutations in the targeted protein or activation of alternative signaling pathways [176]. In addition, the effectiveness of targeted therapy is restricted to a subset of patients with cancer with specific genetic mutations, limiting the utility of targeted therapy across a wide population of individuals with cancer. In this review, we hypothesized that glycolytic enzymes, including GLUT1, HK2, PKM2, PDK1, and LDHA, are promising targets for enhancing the efficacy of EGFR-targeted therapy and overcoming drug resistance in NSCLC. Targeting the glucose metabolism via the Warburg effect has potential advantages over traditional targeted therapies. One advantage of targeting the glucose metabolism in cancer cells is that it may prove to be more effective than traditional targeted therapies for several types of cancers. Although targeted therapy is often specific to certain genetic mutations or cancer types, most cancer cells exhibit a certain degree of the Warburg effect [177]. Another advantage is that targeting the glucose metabolism may result in fewer side effects than those produced by traditional chemotherapies or targeted therapies, because the Warburg effect is a unique characteristic of cancer cells; therefore, drugs targeting this metabolic pathway may be less toxic to healthy cells [79].

PDK inhibitors have demonstrated promising results in overcoming the resistance to EGFR-targeted therapies in NSCLC, suggesting a potential benefit of their use in combination therapies [109,110,111]. Notably, in all studies investigating combination therapies involving PDK inhibitors and EGFR-TKIs to overcome resistance in NSCLC, the PDK inhibitors utilized were of synthetic origin. In a previous study, we discovered that natural product-based PDK inhibitors such as huzhangoside A, Leelamine, and otobaphenol induced PDH activity-dependent cancer cell death [178]. Natural products, shaped by millions of years of evolution, encompass a wide array of chemical compounds exhibiting diverse structures and functions. They stand out as abundant reservoirs of bioactive molecules with potent therapeutic potential [179]. Furthermore, natural products often typically demonstrate superior biocompatibility and reduced toxicity compared to synthetic compounds. This heightened compatibility with biological systems is a result of their natural selection over time [15]. Additionally, natural products often have unique chemical structures that are difficult to replicate using synthetic chemistry, making them valuable sources of novel compounds for drug discovery and development [180]. In light of these compelling attributes, natural product-based PDK inhibitors represent a promising avenue for advancing therapeutic strategies for the treatment of EGFR-TKI-resistant NSCLC.

The upregulation of HIF-1α plays a pivotal role in the adaptation of cancer cells to the tumor microenvironment [181]. Although not a direct glycolytic enzyme, HIF-1α exerts a profound influence on the glucose metabolism by orchestrating the expression of key enzymes and transporters that are involved in glycolysis [181]. In this review, the potential of several natural product-based glycolysis inhibitors that demonstrated the ability to modulate HIF-1α levels was discussed as potential therapeutic strategies for EGFR-TKI-resistant NSCLC (Figure 4). Notably, compounds such as genistein, licochalcone, and sulforaphane exhibited inhibitory effects on HIF-1α, consequently downregulating the expression of crucial glycolytic components including GLUT1, HK2, PKM2, PDK1, and LDHA. Understanding the intricate interplay between HIF-1α and glycolytic pathways is crucial for developing targeted therapeutic approaches, and our findings underscore the promising role of natural product-based glycolysis inhibitors in this context.

Natural products play a pivotal role in cancer therapy, as they offer a vast array of compounds sourced from plants, marine organisms, and microorganisms [182]. These compounds have demonstrated substantial anticancer properties, including the ability to impede cancer cell growth, induce apoptosis, and inhibit angiogenesis [183]. The significance of natural products lies in their unique chemical structures, which often serve as inspiration for the development of novel drugs with improved efficacy and few side effects [183]. Furthermore, natural products exhibit synergistic effects with anticancer drugs and other natural chemicals [184,185].

Some natural product-based glycolysis inhibitors for EGFR-TKI-resistant NSCLC such as apigenin, quercetin, capsaicin, catechin, curcumin, EGCG, leonurine, and sulforaphane have obtained FDA approval for their safety. However, these drugs have been reported to have adverse effects. Capsaicin causes tissue irritation and burning [186]. EGCG, the primary polyphenol in green tea, may manifest side effects such as anxiolytic activity, potential hypoglycemic effects, a risk of hypochromic anemia due to interference with iron absorption, hepatotoxicity, and kidney issues at high doses [187]. High doses of apigenin, particularly from supplements, may cause stomach discomfort, muscle relaxation, and sleepiness [188]. High levels of sulforaphane, catechin, and curcumin can cause digestive difficulties [189,190,191]. Despite FDA approval, these natural glycolysis inhibitors require careful dosing and monitoring because of the potential side effects in therapeutic use. When exploring the pharmacological characteristics of various compounds, factors such as oral bioavailability and water solubility can significantly affect their clinical applicability. Cryptotanshinone and berberine have low oral bioavailability, which limits their clinical applicability [192,193]. Leelamine, α-hederin, and tanshinone IIA demonstrated very low oral bioavailability at 7.6%, 0.14%, and 3.5%, respectively [194,195,196]. Dicoumarol, ursolic acid, genistein, and shikonin face challenges owing to their poor solubility [197,198,199,200]. β-elemene also exhibits poor solubility in water; thus, researchers have synthesized various derivatives to address this issue and enhance its antitumor activities [201]. Licochalcone A, with an oral bioavailability of 3.3% in mice, exhibits poor absorption; however, when loaded onto liposome carriers, its water solubility and oral bioavailability significantly improve [202,203]. Further research is required to explore formulation strategies, (including nanoformulations and lipid-based carriers) to enhance the bioavailability of drugs with poor absorption characteristics and optimize their clinical applicability. More comprehensive research is needed on the pharmacokinetics and pharmacodynamics of hemistepsin A, huzhangoside A, ilimaquinone, and otobaphenol to explore their therapeutic properties. Berberine (NCT03486496), catechin (NCT00573885 and NCT00611650), curcumin (NCT02321293 and NCT01048983), and genistein (NCT01628471 and NCT00769990) are currently undergoing clinical trials for the treatment of NSCLC. Although natural product-based glycolysis inhibitors exhibit diverse drug profile variations in bioavailability, formulation-dependent improvements, and potential side effects, the therapeutic potential of these substances is promising.

In this review, we have presented the potential of combining glycolytic inhibitors with EGFR-TKI to overcome EGFR-TKI resistance. Despite the absence of clinical studies supporting this hypothesis, ongoing research in this field is promising. Further investigation is essential to evaluate the safety, tolerability, and efficacy of the co-administration of glycolytic inhibitors with EGFR-TKIs in a clinical setting. In addition, the effectiveness of glycolytic inhibitors in conjunction with other targeted therapies such as immunotherapy, conventional chemotherapy, and radiotherapy needs to be explored. Furthermore, biomarkers to identify patients who are most likely to benefit from glycolytic inhibitor-based therapies are needed. The identification of biomarkers can create avenues for personalized cancer treatment, thereby enhancing outcomes and mitigating toxicity in patients with cancer. Although the current literature on the utilization of glycolytic inhibitors and EGFR-TKIs is limited, preclinical studies have demonstrated the potential utility of this approach in cancer therapy. As research in this domain progresses, a meticulous evaluation of the safety and efficacy of combination therapy approaches in clinical settings is imperative to enhance the outcomes for patients with cancer.

In conclusion, this review demonstrates that glycolytic inhibitors represent a promising therapeutic strategy for cancer treatment, and that natural products are rich in compounds with glycolytic inhibitory activity. Overall, combining glycolytic inhibitors with EGFR-TKIs can enhance their effectiveness, while minimizing side effects and the risk of resistance.

## Figures and Tables

**Figure 1 ijms-25-00807-f001:**
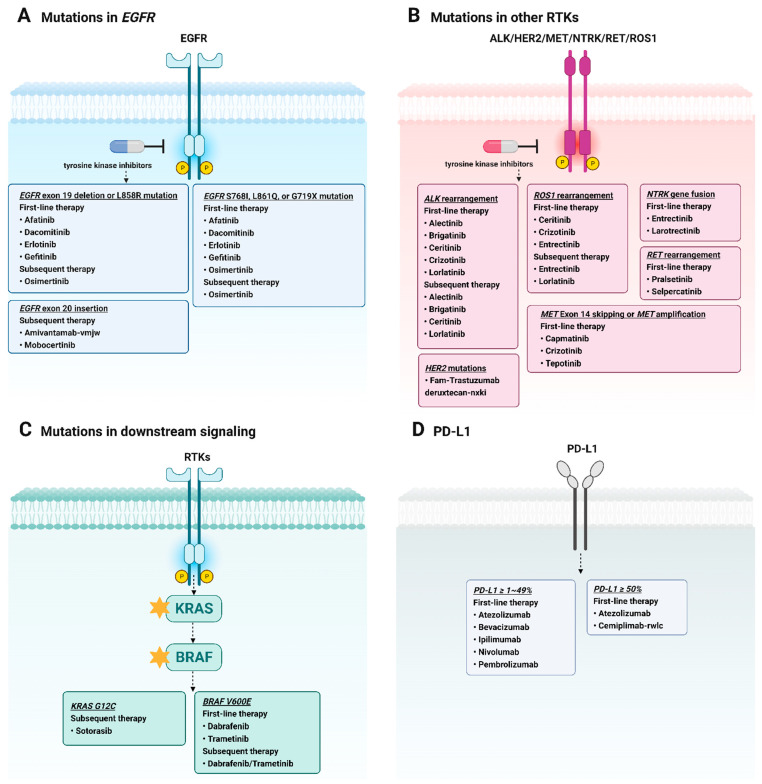
Targeted therapies for lung cancer. Illustration of the current targeted therapies used for non-small cell lung cancer (NSCLC), with specific drugs targeting (**A**) *EGFR* mutations; (**B**) *ALK*, *HER2*, *MET*, *NTRK*, *RET*, and *ROS1* mutations; (**C**) *KRAS* and *BRAF* mutations; and (**D**) immunotherapy drugs for PD-L1 expression. This figure provides an outline of the targeted therapy choices recommended by the 2022 National Comprehensive Cancer Network guidelines for metastatic NSCLC, emphasizing the importance of personalized genetic testing in determining the optimal treatment strategy. *EGFR*, epidermal growth factor receptor; *ALK*, anaplastic lymphoma kinase; *HER2*, human epidermal growth factor receptor 2; *MET*, proto-oncogene, receptor tyrosine kinase; *NTRK*, neurotrophic tyrosine receptor kinase; *RET*, RET proto-oncogene; *ROS1*, ROS proto-oncogene 1, receptor tyrosine kinase; *KRAS*, Kirsten rat sarcoma virus; *BRAF*, v-raf murine sarcoma viral oncogene homolog B1; PD-L1, programmed cell death ligand 1.

**Figure 2 ijms-25-00807-f002:**
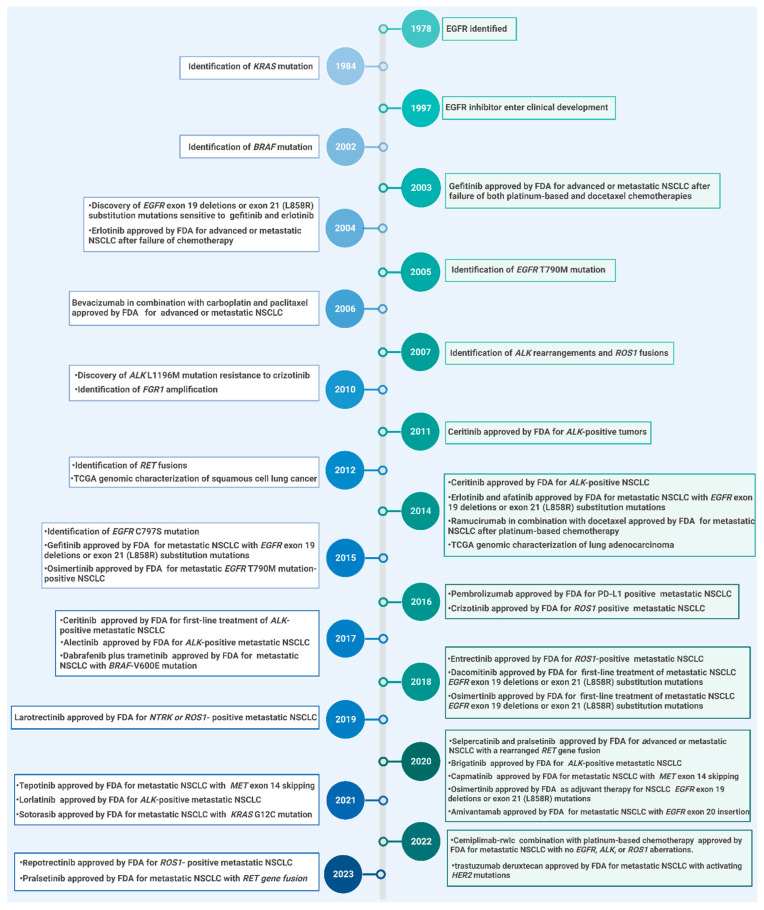
Timeline of non-small cell lung cancer targeted therapy. Illustration of the timeline of genetic alterations in non-small cell lung cancer (NSCLC) subtypes, including *EGFR*, *ALK*, *ROS1*, *KRAS*, *MET*, *PD-L1*, and other mutations. This figure also indicates major concerns regarding the development of targeted therapy for NSCLC. *EGFR*, epidermal growth factor receptor; *ALK*, anaplastic lymphoma kinase; *KRAS*, Kirsten rat sarcoma virus; MET, proto-oncogene, receptor tyrosine kinase; ROS1, ROS proto-oncogene 1, receptor tyrosine kinase; PD-L1, programmed cell death ligand 1.

**Figure 3 ijms-25-00807-f003:**
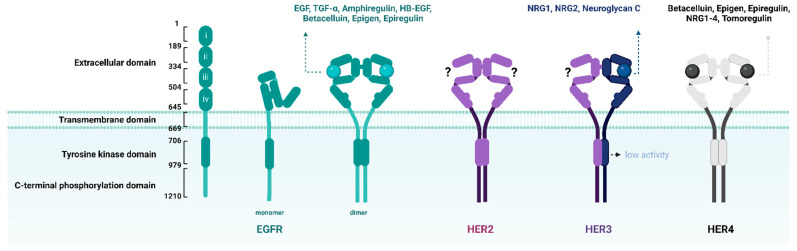
Schematic diagrams of EGFR, HER2, HER3, and HER4. The ErbB protein family includes the EGFR (HER1 and ErbB1), HER2 (Neu and ErbB2), HER3 (ErbB3), and HER4 (ErbB4) proteins. Structurally, EGFR comprises an extracellular domain containing a ligand-binding region, a transmembrane domain, a tyrosine kinase (TK) domain, and a C-terminal phosphorylation domain. Additionally, the binding of growth factors to these receptors is displayed: seven to EGFR, none to HER2, two to HER3, and seven to HER4. Compared with other ErbB protein family members (EGFR, HER2, and HER4), HER3 has little to no TK activity [56,57,58,59]. *EGFR*, epidermal growth factor receptor; *HER*, human epidermal growth factor receptor. ‘?‘ means that ‘None to HER2’ indicates there are presently no known HER2 ligands.

**Figure 4 ijms-25-00807-f004:**
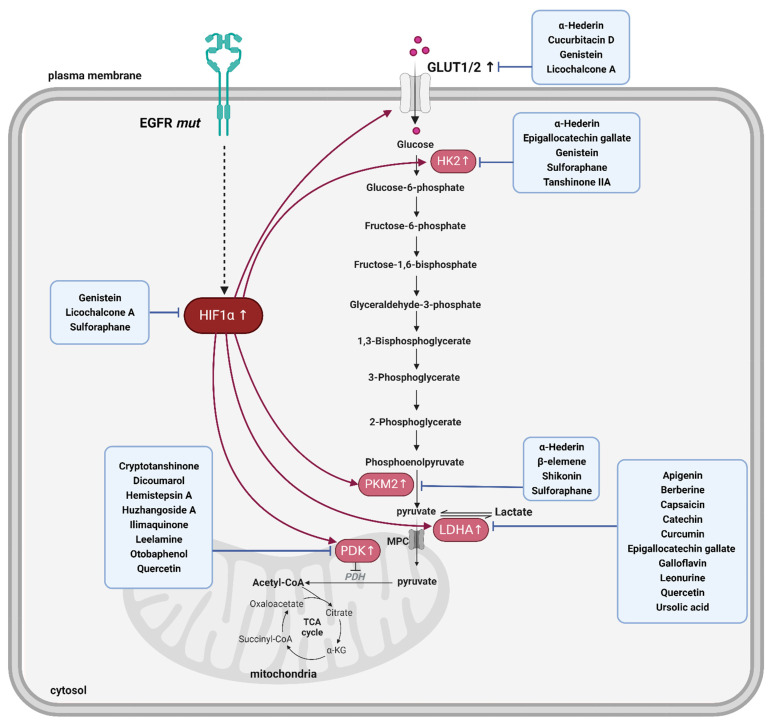
Treatment strategy for *EGFR*-mutated non-small cell lung cancer (NSCLC). Illustration depicting the treatment strategy for NSCLC with *EGFR* mutations. The figure focuses on the glycolysis pathway, a key metabolic process in cancer cells. The glycolysis pathway is highlighted, with key enzymes, including HK2, PKM2, LDHA, and PDK1, marked in pink to emphasize their significance in the metabolic reprogramming of cancer cells. In the blue box, natural product-based glycolysis inhibitors are indicated, showcasing their potential role in targeting glycolytic pathways. Additionally, the figure underscores the regulatory influence of HIF-1α, depicted as a key factor (red), which can upregulate glycolytic enzymes, further emphasizing the intricate interplay within the glycolysis pathway. EGFR, epidermal growth factor receptor; HK2, hexokinase 2; PKM2, pyruvate kinase M2; LDHA, lactate dehydrogenase A; PDK, pyruvate dehydrogenase kinase; HIF-1α, hypoxia-inducible factor 1-alpha. The upward arrow symbol (↑) indicates upregulation.

**Table 1 ijms-25-00807-t001:** Several approved EGFR-TKIs.

Drug	Structure	Drug Type	FDA Approval
Erlotinib (Tarceva^TM^)	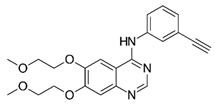	1st-generation EGFR-TKI	1st-line, NSCLC with *EGFR*^19D^/*EGFR*^L858R^ [74]
Gefitinib (Iressa^TM^)	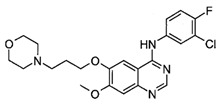	1st-generation EGFR-TKI	1st-line, NSCLC with *EGFR*^19D^/*EGFR*^L858R^ [28]
Afatinib (Gilotrif^TM^)	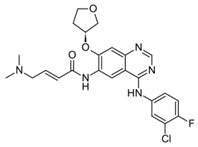	2nd-generation EGFR-TKI	1st-line, NSCLC with *EGFR*^19D^/*EGFR*^L858R^ [25]
Dacomitinib (Vizimpro^TM^)	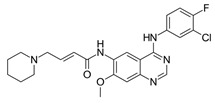	2nd-generation EGFR-TKI	1st-line, NSCLC with *EGFR*^19D^/*EGFR*^L858R^ [26]
Mobocertinib (Exkivity^TM^)	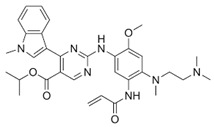	3rd-generation EGFR-TKI	NSCLC with *EGFR* exon20 insertion [75]
Osimertinib (Tagrisso^TM^)	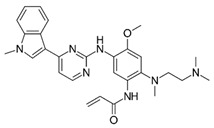	3rd-generation EGFR-TKI	2nd-line, NSCLC with *EGFR*^T790M^ [76] 1st-line, NSCLC with *EGFR*^19D^/*EGFR*^L858R^ [77] adjuvant therapy for NSCLC [78]

**Table 2 ijms-25-00807-t002:** Small-molecule PDK inhibitors derived from natural products.

PDK Inhibitor	Structure	Property	Origin	Clinical Trials for NSCLC	Reference
Cryptotanshinone	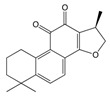	IC_50_: PDK2 (11 µM), PDK4 (>30 µM)	*Salvia miltiorrhiza*	ND	[113]
Dicoumarol	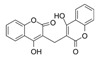	IC_50_: PDK1 (19.42 μM)	*Melilotus officinalis*	ND	[112]
Hemistepsin A	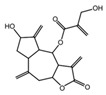	ND	*Hemistepta lyrate*	ND	[18]
Huzhangoside A	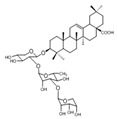	ND	*Anemone rivularis*	ND	[19]
Ilimaquinone	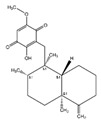	ND	*Smenospongia cerebriformis*	ND	[20]
Leelamine	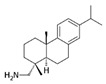	IC_50_: 9.5 µM	bark of pine trees	ND	[120]
Otobaphenol	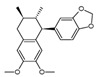	ND	*Myristica fragrans*	ND	[121]
Quercetin	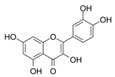	IC_50_: PDK3 (~9.5 μM),	flavonoid glycosides from fruits and vegetables	ND	[122]

ND (not determined).

**Table 3 ijms-25-00807-t003:** LDHA inhibitors from natural products.

LDHA Inhibitor	Structure	Property	Origin	Clinical Trials for NSCLC	Reference
Apigenin	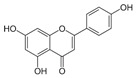	IC_50_: LDHA (0.042 mM)	Flavonoid from fruits, vegetables, and herbs	ND	[139]
Berberine	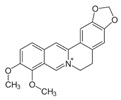	ND	Goldenseal (*Hydrastis canadensis*)	NCT03486496	[140]
Capsaicin	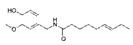	ND	*Capsicum annuum*	ND	[141]
Catechin	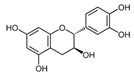	IC_50_: LDHA (40.69 μM)	*Camellia sinensis*	NCT00573885 NCT00611650	[14]
Curcumin	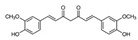	ND	*Curcuma longa*	NCT02321293 NCT01048983	[143]
Epigallocatechin gallate	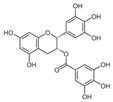	ND	*Camellia sinensis*	ND	[146]
Galloflavin	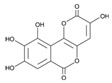	IC_50_: LDHA (5.46 μM)	Flavonoid from food and vegetables	ND	[150]
Leonurine	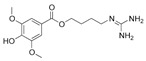	ND	*Leonurus cardiaca*	ND	[151]
Quercetin	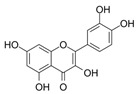	ND	*Quercus*, Flavonoid glycosides from fruits and vegetables	ND	[152]
Ursolic acid	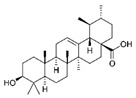	ND	Triterpenoid from citrus fruits and vegetables	ND	[154]

ND (not determined).

**Table 4 ijms-25-00807-t004:** Other glycolysis inhibitors from natural products.

Glycolysis Inhibitor	Structure	Effect	Origin	Clinical Trials for NSCLC	Reference
α-Hederin	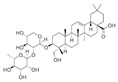	GLUT1, PKM2, LDHA,and HK2↓	*Hedera helix*	ND	[159]
β-elemene	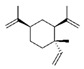	PKM2↓	*Curcuma aromatica*	ND	[160]
Cucurbitacin D	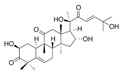	Glut1↓	Cucurbitaceae	ND	[155]
Epigallocatechin gallate	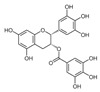	HK2↓	Green tea	ND	[169]
Genistein	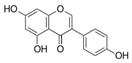	HIF-1α, GLUT1, and HK2↓	Lupin, fava beans, soybeans, kudzu, and psoralea	NCT01628471NCT00769990	[157]
Licochalcone A	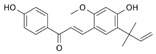	HIF-1α, PDK1, and GLUT1↓	*Glycyrrhiza uralensis*	ND	[162]
Shikonin	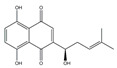	PKM2↓	*lithospermum erythrorhizon*	ND	[174]
Sulforaphane	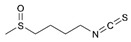	HIF-1α, HK2, and PKM2↓	Broccoli	ND	[166,167]
Tanshinone IIA	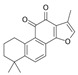	HK2↓	*Salvia miltiorrhiza*	ND	[165]

ND (not determined). The downward arrow (↓) indicates a decrease.

## Data Availability

Not applicable.

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
