# Peer review of "Natural Product-Based Glycolysis Inhibitors as a Therapeutic Strategy for Epidermal Growth Factor Receptor–Tyrosine Kinase Inhibitor-Resistant Non-Small Cell Lung Cancer"

_ijms, 2024, doi:10.3390/ijms25020807_

Round 1
Reviewer 1 Report
Comments and Suggestions for Authors
This is an interesting work. However, some problems were detected:
-abbreviations must be checked in all text. Example: “LDHA” in abstract
-the keywords, timespan, etc of the review must be indicated in the manuscript
-table 1: structure of erlotinib must re redrawn, and TM included in commercial names
-the names of some compounds start with capitals in the middle of sentences, this must be corrected
-the most recent articles in this context should be referred – examples:
https://www.ncbi.nlm.nih.gov/pmc/articles/PMC9748442/
https://aacrjournals.org/mct/article/12/10/2145/91579/Glycolysis-Inhibition-Sensitizes-Non-Small-Cell
https://www.ncbi.nlm.nih.gov/pmc/articles/PMC8342158/
https://jhoonline.biomedcentral.com/articles/10.1186/s13045-022-01311-6
others…
-several relevant compounds were identified; however, their potential drugability, side effects/toxicity should be presented to help researchers towards the development of improved compounds
Comments on the Quality of English LanguageMinor editing of English language required
Reviewer 2 Report
Comments and Suggestions for Authors
Comment 1- Please provide the possible root causes of in one line.
Comment 2- I suggest to provide the name of some FDA approved allopathic drugs used in the treatment of NSLC with their mechanism of action.
Comment 3- Discuss the need of targeted therapy for NSLC.
Comment 4- Careful attention needs to be paid to correct all the grammatical errors and reduce the repetition of the same sentences in paraphrased form in different sections of the manuscript.
Comment 5- Improve the quality of figures.
Comment 6- Expand the ligand of table 1.
Comment 7- In brief, explain following sentence in the light of mechanism involved.
Despite the presence of oxygen, cancer cells often favor glycolysis, a phenomenon known as the Warburg effect [79]. This metabolic adaptation enables cancer cells to efficiently generate energy and biosynthetic precursors essential.
Comment 8- The production of free radicals during glycolysis may have damaging effect on proliferation.
How do the NSLC manage to proliferate in the presence of oxidative stress?
Comment 9- Provide the reference for following statement (Page 7; Line 220)
Targeting key players in the glycolytic pathway, such as GLUT1, 220 HK2, PKM2, PDK, and LDHA, presents a promising avenue for novel therapeutic 221 strategies aimed at overcoming EGFR-TKI resistance.
Comment 10- Write a few lines on the significance of natural products in cancer therapy.
"Hesperidin, a Bioflavonoid in Cancer Therapy: A Review for a Mechanism of Action through the Modulation of Cell Signaling Pathways" Molecules 28, no. 13: 5152. https://doi.org/10.3390/molecules28135152
Comment 11- Is there any clinical studies regarding the investigation of glycolysis inhibitors against NSLC?
If it is there, provide the clinicaltrials govt ID.
Comment 12- Provide a table of abbreviations with their expansions after conclusion section..
In my view, this study is worthy for publication. The manuscript needs minor essential revision before publication.
Comments on the Quality of English Language
Careful attention needs to be paid to correct all the grammatical errors
Round 2
Reviewer 1 Report
Comments and Suggestions for Authors
The manuscript was improved and now it is more acceptable for publication.